# Identification of Atypical Circulating Tumor Cells with Prognostic Value in Metastatic Breast Cancer Patients

**DOI:** 10.3390/cancers14040932

**Published:** 2022-02-13

**Authors:** Alexia Lopresti, Claire Acquaviva, Laurys Boudin, Pascal Finetti, Séverine Garnier, Anaïs Aulas, Maria Lucia Liberatoscioli, Olivier Cabaud, Arnaud Guille, Alexandre de Nonneville, Quentin Da Costa, Emilie Denicolai, Jihane Pakradouni, Anthony Goncalves, Daniel Birnbaum, François Bertucci, Emilie Mamessier

**Affiliations:** 1Laboratory of Predictive Oncology, Cancer Research Center of Marseille, Inserm U1068—CNRS UMR7258—Université Aix-Marseille UM105, Label “Ligue Contre le Cancer”, 13009 Marseille, France; alexia.lopresti@gmail.com (A.L.); claire.acquaviva@inserm.fr (C.A.); laurys.boudin@intradef.gouv.fr (L.B.); finettip@ipc.unicancer.fr (P.F.); seve.garnier@gmail.com (S.G.); anais.aulas@inserm.fr (A.A.); maria-lucia.liberatoscioli@inserm.fr (M.L.L.); olivier.cabaud@inserm.fr (O.C.); arnaud.guille@inserm.fr (A.G.); alexandre.tassin-de-nonneville@inserm.fr (A.d.N.); quentin.da-costa@inserm.fr (Q.D.C.); denicolaie@ipc.unicancer.fr (E.D.); daniel.birnbaum@inserm.fr (D.B.); bertuccif@ipc.unicancer.fr (F.B.); 2Sponsor Unit, Department of Clinical Research and Innovation, Institut Paoli-Calmettes, 13009 Marseille, France; pakradounij@ipc.unicancer.fr; 3Department of Medical Oncology, Institut Paoli-Calmettes, 13009 Marseille, France; goncalvesa@ipc.unicancer.fr

**Keywords:** circulating tumor cells, breast cancer, metastases, epithelial-to-mesenchymal transition, plasticity, biomarker, cluster, CTC, CTM, survival

## Abstract

**Simple Summary:**

In this study we have isolated and analyzed atypical cells found in the blood of metastatic breast cancer patients using a micro-filtration technic. This technic, being very easy to implement, was also extremely useful for studying circulating tumors cells’ (CTCs) heterogeneity in cancer patients. We highlighted three subsets of CTCs, with different independent unfavorable prognostic values for progression-free and overall survival. We demonstrated that these cells can further be analyzed by immunofluorescence to narrow their molecular profiles and identify specific characteristics. Moreover, we identified a subset of CTCs, for which positivity might be a useful stratification tool to select patients more susceptible to benefit from early clinical trials testing novel therapeutics, which frequently enroll late-stage, already heavily pre-treated and thus poor-responder patients.

**Abstract:**

Circulating tumor cells have a strong potential as a quasi-non-invasive tool for setting up a precision medicine strategy for cancer patients. Using a second-generation “filtration-based” technology to isolate CTCs, the Screencell™ technology (Sarcelles, France), we performed a large and simultaneous analysis of all atypical circulating tumor cells (*aCTCs*) isolated from the blood of metastatic breast cancer (mBC) patients. We correlated their presence with clinicopathological and survival data. We included 91 mBC patients from the PERMED-01 study. The median number of *aCTCs* was 8.3 per mL of blood. Three subsets of *aCTCs*, absent from controls, were observed in patients: single (*s-aCTCs*), circulating tumor micro-emboli (*CTM*), and giant-aCTCs (*g-aCTCs*). The presence of *g-aCTCs* was associated with shorter progression free survival and overall survival. This study highlights the heterogeneity of *aCTCs* in mBC patients both at the cytomorphological and molecular levels. In addition, it suggests the usefulness of the *g-aCTC* subset as a prognostic factor and a potential stratification tool to treat late-stage mBC patients and improve their chances of benefiting from early clinical trials.

## 1. Introduction

Metastases are responsible for more than 90% of cancer-associated mortality. Anticipating and understanding their evolution using simple and reliable biomarkers is paramount. Some tumor cells, shed from primary or metastatic tumors, can enter the blood circulation and, in this circumstance, are specifically called “Circulating Tumor Cells” (CTCs). A few of them, fit enough to survive the drastic conditions encountered in the blood flow (anoikis, sheer stress, anti-tumor immunity, etc.), will eventually spread to other organs and might initiate novel malignant foyers [1].

CTC sampling is minimally invasive and can be repeated on demand. It represents an “*easy*” access to distant tumor lesions. In the early 2000s, the Cellsearch™ technology (Menarini–Silicon Biosystems, Florence, Italy) was FDA-approved to count CTCs. Thanks to this technology, CTC enumeration at diagnosis is now recognized as an independent predictor of progression-free survival (PFS) and overall survival (OS) in metastatic solid tumors such as breast, colon, and prostate cancers [2,3]. In metastatic breast cancer (mBC) patients, variation in CTC counts during chemotherapy is predictive for disease progression and survival on a real-time basis, as therapy proceeds [3,4]. Nonetheless, CTCs’ prognostic value has a poor added clinical utility in metastatic patients and remains unproven in non-metastatic patients [2,5].

Second-generation technologies for the detection of CTCs, based on cell size and deformability, revealed a surprising layer of complexity. They showed that CTCs detected with the CellSearch™ represent only a small fraction of a more heterogeneous pool of CTCs. Based on cytomorphological characteristics, cells with malignant morphological features found in the blood of cancer patients, but not in healthy donors, will be referred to as an “*atypical Circulating Tumor Cell*” or *aCTC*. The current hypothesis is that the total count of *aCTCs* might lower their prognostic value because of the intrinsic biological heterogeneity of this population [6]. Indeed, data from small cohorts of patients with solid tumors have reported the presence of at least three totally different subsets of *aCTCs* [7,8,9]. They were named after their main cytological presentation: single-atypical CTC (*s-aCTC*), circulating tumor micro-emboli (*CTM*), and giant-atypical CTC (*g-aCTC*). 

Here, we have studied the heterogeneity and the prognostic value of *aCTCs* subset in mBC refractory to systemic therapy. For this, we used the Screencell^®^Cyto device (ScreenCell™, Sarcelles, France), one of the most promising and easy-to-use filtration-based technologies. 

## 2. Results

### 2.1. Patients’ Population

Ninety-one mBC patients enrolled in the PERMED-01 trial were included. Patients’ median age at inclusion was 55 years (range, 27–79) (Table 1). The median time between diagnosis of metastatic relapse and primary cancer was 3 years (range, 0.6–23). Regarding the primary tumor, the most frequent pathological type was ductal (93%), the most frequent pathological grade was 3 (50%), and the molecular subtypes were mainly HR+/HER2− (56%), then triple-negative (TN) (32%), and HER2+ (12%). The molecular subtype of the metastatic biopsies, available for 87 out of 91 patients, included more TN cases (46%) than HR+/HER2− (41%) and HER2+ (13%). Nineteen patients (22%) showed discordance between the molecular subtype observed in the metastatic sample and the primary tumor. The median number of metastatic sites at inclusion was 3 (range, 1–8). The most frequent sites were bone (65%), lymph node (63%), and liver (57%). The median number of lines of systemic treatments received before inclusion was 4 (range, 1–10). Regarding chemotherapy, the median number of previous lines received after inclusion was 3 (range, 1–7). As expected, 91% of patients had previously received taxane and/or anthracycline treatments. The systemic treatments received before and after inclusion are summarized in Appendix A. With a median follow-up of 12 months after inclusion (range, 1–52), all but one patient showed disease progression and 68 died. The 1-year PFS was 13% (95% CI 8–22), the median PFS was 5 months (range, 1–47), 1-year OS was 58% (95% CI 40–70), and the median OS was 14 months (range, 1–52). 

### 2.2. Three Subsets of aCTCs Are Found in the Blood of mBC Patients

Very few data provide simultaneous characterization of the three subsets of *aCTC* previously found in the blood of cancer patients. We wondered whether an identification based on cytomorphological criteria, independent from markers, as described in [7,10], could be a quick way to identify *aCTCs* subsets in mBC patients. We thus screened 91 blood samples from advanced mBC for the presence of *aCTCs* after MGG coloration of a ScreenCell^®^Cyto filter. We observed that three subsets of *aCTCs*, previously described in other solid tumors and on some rare occasions in mBC, were present in the blood of mBC patients. These *aCTCs* present themselves as single cells (*s-aCTCs)* (Figure 1a), clusters of cells (*CTM)* (Figure 1b and Appendix A), or giant cells (*g-aCTCs)* (Figure 1c).

### 2.3. The Count of aCTC Subsets Is Highly Variable in mBC Patients

Next, we characterized the frequency and distribution of each subset individually and for each patient (Figure 1d, Appendix A). All but four patients (96%) had at least one *aCTC* per mL of blood. The median number of *aCTCs* per mL was 8.33 (range, 0–481.6). The median number of *s-aCTCs* per mL was 2 (range, 0–51), with 72 patients with at least one *s-aCTC* per mL (79%). The median number of *CTM* was 1.33 per mL (range, 0–479.6), with 47 patients with at least one *CTM* per mL (52%). The median number of *g-aCTC* per mL was 0 (range, 0–10.3), with 33 patients with at least one *g-aCTC* per mL (36%). The distribution of the different cell subsets was variable between patients: *s-aCTCs* were more frequent than *CTM*, which were more frequent than *g-aCTCs.* When present, the *CTM* and *s-aCTCs* were in larger concentration per mL of blood than were *g-aCTCs*. 

Based on the distribution of each *aCTC* subset, we defined a positivity cut-off using a two-component Gaussian finite Mixture Model to establish correlations between *aCTCs* subsets and clinicopathological data. Above the cut-off, patients were considered “positive” for this cell subset. The number of positive patients was 41 (45%) for all *aCTCs* (cut-off: >4.5 cells/mL), 49 (54%) for *s-aCTCs* (cut-off: ≥1.8 cells/mL), 22 (24%) for *CTM* (cut-off: ≥0.9 cells/mL), and 42 (46%) for *g-aCTCs* (cut-off: ≥0.33 cells/mL) (Appendix A).

### 2.4. Different aCTC Subsets Correlate with Different Clinicopathological Features

Correlations between *aCTCs* or each individual subset cut-off values and clinicopathological features of patients and tumors (logit function test) are listed in Appendix A. Even though these results must be interpreted with caution given the high number of statistical tests done for a medium-size cohort, interesting correlations were nonetheless observed. A number of *aCTCs* above the cut-off was associated with the presence of liver metastases (*p* = 3.13 × 10^−2^) and other *aCTC* subsets above their cut-offs. 

The three cell subsets showed different correlations (Figure 2a). The presence of *s-aCTCs* was more frequently associated with the HR+/HER2− subtype of primary cancer (*p* = 1.29 × 10^−2^) and of metastatic lesion (*p* = 3.22 × 10^−2^), with the presence of liver metastases (*p* = 4.56 × 10^−2^), the absence of skin metastases (*p* = 2.86 × 10^−2^), and the positivity of *g-aCTCs* (*p* = 2.29 × 10^−3^). The presence of *CTM* was associated with a shorter time between metastatic relapse and diagnosis of primary cancer (*p* = 4.51 × 10^−2^) and the TN vs. HR+/HER2− subtype of primary cancer (*p* = 3.39 × 10^−2^). The presence of *g-aCTCs* was associated with the existence of peritoneal metastases (*p* = 3.75 × 10^−2^) and the positivity of *s-aCTCs* (*p* = 2.29 × 10^−3^).

None of the *aCTC* subsets was associated with patients’ age, number of metastatic sites and number of previous lines of chemotherapy at inclusion, or pathological type and grade of primary tumor.

### 2.5. The g-aCTCs “Positive” Status Is an Independent Prognostic Factor for PFS and OS

We then asked if all *aCTCs* or individual *aCTC* subsets were associated with patients’ survival. We did uni- and multivariate analyses (Table 2). In univariate analysis, patients with numbers of *aCTCs* above the positivity cut-off (“positive”) displayed shorter PFS than “negative” patients (HR = 1.37, 95% CI 0.89–2.10), but the difference was not significant (*p* = 0.158). Such association was also not significant for *s-aCTCs* and for *CTM*. By contrast, it was significant for the *g-aCTCs* subset, with a HR for PFS event of 1.94 (95% CI 1.24–3.01) in the “positive” patients as compared to the “negative” patients (*p* = 2.98 × 10^−3^). The 1-year PFS was 5% (95% CI 1–18) in the *g-aCTC* “positive” vs. 20% (95% CI 12–35) in the “negative” group (*p* = 2.46 × 10^−3^; Figure 2b), and the respective median PFS were 3.6 months (range, 3–4.3) and 5.3 months (range, 4.8–7.4). The other clinicopathological features associated with PFS in univariate analysis included the patients’ age at inclusion (*p* = 3.34 × 10^−2^) and the number of previous lines of chemotherapy at inclusion (*p* = 3.48 × 10^−2^). In multivariate analysis, the *g-aCTC* status remained significantly associated with shorter PFS (*p* = 2.19 × 10^−2^), suggesting independent prognostic value.

Similar results were observed with OS. In univariate analysis, the patients’ status for all *aCTCs*, *s-aCTCs*, and *CTM* was not significantly associated with OS (Table 2). By contrast, it was significant regarding *g-aCTCs* with an HR for death of 2.46 (95% CI 1.47–4.12) in the “positive” vs. “negative” patients (*p* = 5.84 × 10^−4^). As shown in Figure 2c, the 1-year OS was 71% (95% CI 59–85) in the “*g-aCTC*-negative” group vs. 42 (95% CI 29–62) in the “*g-aCTC*-positive” group (*p* = 1.06 × 10^−4^, log-rank test). The respective median OS were 9.2 months (range, 6.1–13.9) and 18.7 months (range 14.5–24.8). The other variables associated with OS in univariate analysis included the presence of brain/meningeal metastases (*p* = 1.15 × 10^−3^) and the number of previous lines of systemic therapy at inclusion (*p* = 5.84 × 10^−4^). Again, in multivariate analysis, the *g-aCTC* status remained significant and associated with shorter OS (*p* = 2.98 × 10^−3^), suggesting independent prognostic value.

### 2.6. A hybrid Epithelial–Mesenchymal Phenotype Is Associated with LGR5 and ABCB1 Markers Co-Expression

Because each *aCTC* subset correlated with different clinico-pathological features and had different prognostic values, we wondered whether each subset might display distinct phenotypes. We analyzed *aCTC* subsets more in-depth by looking at their EMT status (EPCAM, Pan-KRT, and VIM) and the expression of LGR5 and ABCB1 markers. These markers are often modulated in cells with a mesenchymal-biased profile [11,12,13,14], with stemness attributes [15] or resistance to systemic treatments [16]. This list of markers is of course not exhaustive but aims at highlighting potential functional traits observed in progressing diseases, a characteristic of our cohort of patients. The *aCTCs* from the last 23 patients of the cohort were analyzed using advanced multicolor confocal microscopy. This analysis involved a total of 2152 *aCTCs* (*n* = 336 *s-aCTCs, n* = 1742 *CTM*, and *n* = 74 *g-aCTCs*). The data obtained with the simultaneous combination of these markers, for each *aCTC* subset, are unprecedented (Figure 3a and Appendix A).

An epithelial-strict (EPCAM+ and/or KRT+ but VIM-) phenotype was found on very few *aCTCs*, independently of the subset of origin (2.6% of *s-aCTCs*, 0.6% of *CTM,* and 0% of *g-aCTCs*) (Figure 3b and Appendix A). The predominant phenotype of *s-aCTCs* was mesenchymal (≈70%). The rest of the *s-aCTCs* (27%) displayed a partial EMT, also called hybrid Epithelial–Mesenchymal (E/M) phenotype. The majority of cells observed in *CTM* and *g-aCTCs* displayed a hybrid E/M phenotype (82.5% and 63.6%, respectively); few *CTM* and *g-aCTCs* were mesenchymal-only (16.4% and 27.4%, respectively). Altogether, these results show that a hybrid E/M phenotype, and to a lesser extent a mesenchymal phenotype, were the most common phenotype observed in *aCTCs* (Figure 3b). We also looked at LGR5 and ABCB1 markers expression. The *s-aCTCs* did not show any preferential pattern for LGR5 or ABCB1 expression (Appendix A). More than 80% of cells in *CTM* expressed LGR5 and 65% of them were positive for both LGR5 and ABCB1. Finally, 90% of *g-aCTCs* co-expressed LGR5 and ABCB1, independently of their EMT status (Figure 3b, Appendix A). Altogether, and even if these data are preliminary and require additional tests to confirm the functional status of each subset, they highlight the phenotypic heterogeneity that parallels the cytomorphological heterogeneity of *aCTCs*.

### 2.7. A hybrid Epithelial–Mesenchymal Phenotype Is Associated with Shorter PFS

Numerous studies show that cells with a hybrid E/M phenotype have a higher metastatic potential than other cells due to enhanced survival abilities [17]. In an exploratory analysis, we assessed the prognostic value of *aCTCs’* EMT status in the small subgroup of 23 patients. This status was discretized using a 50% positivity cut-off and was considered a positive when more than 50% of *aCTCs* displayed a hybrid E/M phenotype. In univariate analysis (Appendix A), the hybrid E/M phenotype was associated with shorter PFS (HR = 2.87, 95% CI 1.08–7.68; *p* = 3.53 × 10^−2^). As shown in Appendix A, the patients without *g-aCTCs* and without *aCTCs* of hybrid E/M status (*g-aCTC* 0 + E/M 0) displayed a 6-month PFS of 71% (95% CI 45–100) vs. 17% (95% CI 3–100) for patients with both positive status (*g-aCTC* 1 + E/M 1) and 40% (95% CI 19–85) for patients not matching these classes (*g-aCTC* 0 + E/M 1 and *g-aCTC* 1 + EM0) (*p* = 1.12 × 10^−2^, log-rank test). Such prognostic complementarity was tested using the likelihood ratio (LR) test: the *g-aCTC* status added prognostic information to that provided by the hybrid E/M phenotype of *aCTCs* (ΔLR−χ^2^ = 4.08, *p* = 4.34 × 10^−2^; Appendix A), and, conversely, this latter added prognostic information to that provided by the *g-aCTC* status alone (ΔLR−χ^2^ = 2.89, *p* = 8.9 × 10^−2^). This preliminary result shows that investigating both the molecular phenotype of *aCTCs* and their cytological aspect could improve the prognostic accuracy of *aCTCs*. Given the very small number of patients, this result must be considered as hypothesis-generating only.

## 3. Discussion

The ScreenCell^®^Cyto device allowed the cytomorphological identification of at least three cellular subsets of *aCTCs* in the blood of mBC patients: *s-aCTCs*, *CTM,* and *g-aCTCs*. Each subset showed correlation with various clinical variables, but only the *g-aCTCs* had an independent prognostic value for survival.

### 3.1. Reliability

Single-use filter-based systems provide a strong efficiency in isolating *aCTCs*, while most erythrocytes and leukocytes pass through. Compared to other tools, recent data demonstrate that this single-use filter-based device has one of the best sensitivity/specificity balances to isolate *aCTCs* from the blood of patients with solid tumors [8,18,19]. Size-based CTC separation methods, which rely on cell size and deformability, a parameter that is strongly affected by EMT, might however be biased toward enrichment of mesenchymal *aCTCs* [20]. 

### 3.2. Comparison

The cytological subsets described here have been reported elsewhere. It is of note that these data concern metastatic patients and remain particularly rare in non-metastatic patients [21]. Studies that simultaneously looked at the three subsets in mBC are also scarce, except for one study that is more technical than clinical [22]. In this study *(n* = *20)*, the prevalence of *single*, *CTM,* and *giant* CTCs was, respectively, of 67%, 27%, and 75% compared to 79%, 52%, and 46% in our cohort (*n* = 91). Major discrepancies can be noted regarding the prevalence of *CTM* and *g-aCTC*, respectively, with a higher and lower prevalence in our study.

Concerning the *CTM*, we verified that clusters of more than three cells were not artificially formed during the filtration process (Appendix A). This control and the larger size of the patients’ cohort compared to other studies are in favor of a high prevalence of *CTM* in advanced mBC. Although their prevalence is lower than that of *s-aCTCs*, *CTM* have an increased metastatic potential in BC [23,24]. We found that *CTM* predominantly have a hybrid E/M phenotype and express the LGR5 marker. This has been observed in tumors with high TNM stage [25]. The hybrid E/M phenotype combines the advantages of the mesenchymal phenotype, which confers increased invasiveness and drug resistance to cancer cells, and the epithelial phenotype, endowed with higher proliferation abilities and seeding capacities [26]. In mBC, more *CTM* release and a dynamic shift toward a hybrid E/M phenotype during treatment were correlated with treatment failure and disease progression [27]. This is coherent with our results. We were, however, not able to correlate *CTM* presence to shorter patients’ PFS or OS. Given that our cohort had already received multiple prior systemic therapies, one can hypothesize that the predictive value is less powerful in terminal disease. Analyses of variations between time points might be more informative [28,29]. Nonetheless, the high prevalence of *CTM* with a hybrid E/M phenotype is an important and new observation in patients with advanced mBC not responding to treatment. This suggests that persistent *CTM* could be an important prognostic marker during treatment.

Explanations regarding the discrepancy in *g-aCTC* prevalence could be multiple. The prevalence of these giant cells in the blood of patients with a solid tumor is around 40% [30,31,32]. Discrepancy might be due to the greater volume of blood tested (7.5 vs. 3 mL in our case), which increases the chance of detecting rare cells.

Most intriguing is the nature of these giant cells, which have been identified in patients with cancers but not in healthy donors [33]. They were given different names: circulating hybrid E/M cells, giant epithelioid cells, fusion-derived epithelial cancer cells, tumor-macrophage fusion cells (TMFs), macrophage-tumor cell fusion cells (MTFs), or cancer-associated macrophage-like (CAMLs) cells [22,30,32,33,34,35,36]. This shows that the origin of *g*-*aCTCs* is unclear. Their formation may be consecutive to cellular fusion of tumor cells with other tumor cells (homotypic) or with non-tumorous cells (heterotypic). The fusion’s product supposedly combines the attributes of both fused cells and results in a cancer cell with a more aggressive phenotype, responsible for shorter survival and increased metastatic and chemoresistance capabilities. Similarly, artificial fusion of tumor cells with macrophages shows enhanced migratory, invasive, and metastatic phenotypes in vivo [30,32,37]. It is interesting to note that if *g-aCTCs* are the result of cell fusion, their potential polyploidy could enhance their chances to adapt to environmental changes. This would increase the tumorigenic potential of *g-aCTCs* compared to other subsets of *aCTCs*. *g-aCTCs* polyploidy may become an interesting marker for cell plasticity. 

Other studies have identified these large cells as cells related to the macrophage lineage, describing them in a broad way as giant macrophages (CAMLs, MTF, or MFT) that contain phagocytosed epithelial debris [22,31,33,34,35,38]. These cells indeed express CD14 or CD11c, two markers of the macrophage/myeloid lineage. Both hypotheses, fusion or myeloid origin, require further investigations to clarify the origin and function of these giant cells. This would help refine their role as biomarkers, including in early disease [38].

A seminal study revealed that the type of treatment received (hormone-treated vs. chemotherapy) affects the quantity of CAMLs detected [33]. It suggested that CAMLs may provide a sensitive representation of phagocytosis of cellular debris caused by chemotherapy at the tumor site. In our population, treatments, including chemotherapy, were inefficient. It is thus plausible that CAMLs or *g-aCTCs,* if we assume the overlap between these two populations, are less frequently released because of cancer cells’ resistance to treatment at metastatic sites. 

ABCB1 efflux pump is involved in the resistance to a number of anticancer agents used for mBC treatment, including anthracyclines, vinca alkaloids, and taxanes, among others [16]. Here, we have shown its expression in g-*aCTC* and *CTM* subsets. If ABCB1 expression is proved responsible for resistance to chemotherapeutic agents, the combined detection of *aCTCs* and ABCB1 marker might be extremely valuable to estimate the efficiency of a given treatment as the therapy proceeds. However, the molecular part of our study in 23 patients is still very preliminary and warrants future validation in larger series. For now, it should be regarded only as hypothesis-generating.

### 3.3. Limitation

The major limitation of our study was to rely on cytomorphological criteria to detect *a-CTCs,* although based on stringent criteria used in clinical practice. Studies including additional markers to determine *a-CTCs* subsets phenotype and importance are still warranted.

Another obvious weakness is the small number of samples compared to the large range of prior systemic therapies and the heterogeneity in term of molecular subtypes, which likely explains the non-significant prognostic value of tumor grade and molecular subtypes of our cohort. However, our analysis included the best validated prognostic factors [39], such as relapse-free interval, site of metastases, number of metastatic sites and molecular subtypes. Ideally, similar analyses should be reapplied to larger and more homogeneous series of patients.

Finally, the high sensitivity of the ScreenCell^®^Cyto (ScreenCell™, Sarcelles, France) device might result in the trapping of non-malignant cells on the filters [40]. Endothelial cells, although not supposedly found circulating in the blood, have been identified in rare occasions in clusters, notably in cancer patients [41,42]. Against all expectations, some of these endothelial clusters displayed cytological traits consistent with malignancy, such as atypical nuclei, prominent nucleoli, and a high nuclear-to-cytoplasmic ratio. However, in mBC patients, the analysis of their cell-type inference revealed an epithelial-derived tumor cell origin [24]. 

### 3.4. Benefit and Practical Implication

Our work presents the largest cohort of mBC patients analyzed so far using the ScreenCell^®^Cyto system. It comforts the validity and utility of this easy, relatively inexpensive, and extremely convenient system, to study *aCTCs*. Still, it requires the expertise of a cytologist for analysis. In this prospect, the development of an automatized detection of *aCTC* subsets is ongoing (Project Medicen, part of the NCT03797053 trial). Validation has already been achieved with samples from melanoma patients, showing a sensitivity of automated counting compared to the conventional reading of 97% (11th International Symposium on Minimal Residual Cancer, May 2018).

## 4. Materials and Methods

### 4.1. PERMED-01 Sub-Study

The PERMED-01 study was a prospective monocentric clinical trial, promoted by and conducted at the Paoli–Calmettes Institute (Marseille, France) and registered as identifier NCT02342158 at the ClinicalTrials.gov platform. More details are available in the Appendix A. The present work is an ancillary study of the PERMED-01 assay and is oriented on the analysis of aCTC subsets isolated from the blood of patients with mBC refractory to at least one line of systemic therapy and with an accessible lesion to biopsy at the time of inclusion [43]. The trial was approved by the French National Agency for Medicine and Health Products Safety, a national ethics committee (CPP Sud-Méditerranée) and our Institutional Review Board. It was conducted in accordance with the Good Clinical Practice guidelines of the International Conference on Harmonization. All patients gave their informed consent for inclusion, biopsy and blood sampling, and molecular analysis. A total of 91 adult female patients, enrolled between January 2015 and December 2016, were sampled for aCTC analysis. For each subject, a blood sample of 5 mL was collected in a Vacutainer^®^ tube containing EDTA K2. The first milliliters of blood were discarded to avoid contamination of the endothelial cells during the puncture. All samples were shipped at 4 °C to the laboratory and processed extemporaneously upon reception within 4 h.

### 4.2. Atypical Circulating Cells Enrichment Using ScreenCell^®^CYTO Device

The Screencell^®^Cyto device (ScreenCell™, Sarcelles, France) is an approved CE-IVD test. Blood samples were processed with this device, as described elsewhere [7,10]. In brief, 3 mL of peripheral blood were incubated with a red blood cell lysis/fixative buffer for 8 min at room temperature (RT). A low pressure allows the suspension to pass through a metal-rimmed filter, dotted with 7.5 ± 0.4 µm diameter pores. The metal-rimmed filter was rinsed with Phosphate Buffered Saline (PBS), dislodged from the device, then dried at RT before use with May Grünwald Giemsa (MGG) and/or immunofluorescence staining.

### 4.3. Atypical Circulating Cells Staining with May Grünwald Giemsa (MGG)

The filters were stained using an MGG kit (Merck Millipore, Molsheim, France) according to the manufacturer’s instructions. Afterwards, filters were air-dried at RT for 5 min and stored protected from light until subsequent analysis by light microscopy using a Leica™ microsystem light microscope (10×, 20× and oil-40× objectives) and the NIS-Elements Viewer software (Nikon Instruments Inc.,Melville, NY, USA). The complete screening of the metal-rimmed filter, which can be performed retrospectively, takes 10 to 15 min for an experienced eye.

### 4.4. Atypical Circulating Cells Subsets according to Cytological Criteria by MGG

To avoid biases in the comparative analysis of samples, the screening of processed filters was done by two investigators (EM and AL). Every potential *aCTC* was documented and categorized by the trained cytologists, as previously described [7]. Correlation between cytologists was above 92% concordance, and questionable interpretations were selected for discussion until a consensus was reached. In very rare occasions, the event was considered as “uncertain” and thus excluded from the analysis (Appendix A). 

Potential *aCTCs* were identified using the following criteria: cell size and shape, irregularity of nuclear borders, enlarged nucleus size, anisonucleosis, dense hyperchromatic nucleus, not totally opaque, nucleocytoplasmic (N/C) ratio [7,44,45,46]. Cells without visible cytoplasm were not included in this study. Details about the criteria used for each subset are listed in Figure 1a–c and in Appendix A. Briefly, *s-aCTCs* were identified as epithelioid cells with enlarged irregular hyperchromatic nuclei (≈20 ± 4 μm) and a high nucleocytoplasmic (N/C) ratio (>0.75). CTM identification criteria were based on the cluster of at least three cells, with nuclei showing signs of anisonucleosis, often with irregularity of nuclear borders and dense hyperchromatic nuclei [10,22,45]. The *g-aCTCs* were very large individualized cells (≈50–300 µm) identified based on a voluminous cytoplasm (low N/C ratio), which can be round or oblong, and with an enlarged nuclear profile (>20 μm in diameter), often multilobular or with separated polymorphic nuclei [33,47].

### 4.5. Immunofluorescence Staining of aCTCs for Confocal Microscopic Analysis

We set up a 6-color immunofluorescence staining simultaneously targeting leukocytes (CD45), epithelial markers (EPCAM and Pan-cytokeratin: pan-KRT), mesenchymal marker (VIM), the stem cell marker LGR5, and efflux pump ABCB1 and SytoxBlue as a DNA labeling dye. The antibodies are detailed in Appendix A. All antibodies used in the combination were first validated on cell lines with known positive or negative expression for these markers. The antibodies’ specificity, signal/noise ratio, and antibodies combination are shown in Appendix A.

Immunofluorescence staining was done on 23 filters, fixed in paraformaldehyde 4% for 5 min at RT, rinsed 3 times in PBS, dried then stored at 4 °C until use (stopping point). Filters were then re-hydrated in TBS and permeabilized in TBS-0.2% Triton-X100 for 5 min on the day of use. After a quick rinse in water, the filter was incubated in blocking buffer (3% Bovine Serum Albumin, 1% Donkey serum, and 1% Goat serum) at RT for 30 min to minimize non-specific staining. Primary antibodies were added in blocking buffer and incubated overnight at 4 °C. Following three washes with TBS 0.05% Tween20 (TT20), the secondary antibody mixture was added for 1 h in the dark at RT. After three washes with TT20, the anti-CD45-A488 antibody was incubated for 1 h in the dark at RT. Finally, after three washes with TT20 and water, the filter was counterstained with Sytox Blue Nucleic Acid Stain for 5 min at RT in the dark (Life Technologies SAS). Finally, the filters were mounted with Kaiser Solution (Sigma Aldrich, MERCK) and dried at RT a few hours. Immunofluorescence was analyzed with a c-Plan-Apochromat 40×/1.3 oil objective on a LSM880 confocal with spectral detection from Zeiss equipped with a 405-laser diode, an Argon-laser, and 561- and 633-lasers. The images’ acquisition and spectra unmixing were conducted using the Zen Black software (Zeiss).

### 4.6. Statistical Analysis

Correlations between aCTC subsets and clinicopathological variables were established using logistic regression (Logit link function). PFS and OS were calculated from the date of inclusion in PERMED-01 until the date of first progression or death from any cause. Follow-up was measured from the date of inclusion to the date of last news for event-free patients. Survivals were calculated using the Kaplan–Meier method and curves were compared with the log-rank test. Uni- and multivariate prognostic analyses were performed using Cox regression analysis (Wald test). The variables submitted to univariate analyses included patients’ age at inclusion, metastasis-to-diagnosis time, pathological type, grade and molecular subtype of primary tumor, molecular subtype of metastasis biopsied in PERMED-01, nature and number of metastatic sites at inclusion, number of previous lines of chemotherapy at inclusion, and *aCTC* subsets. Multivariate analyses included the variables significant in univariate analysis (*p* ≤ 0.05). The cell subset discretization was based on the distribution of count cell values of each *aCTC* subset independently, and cut-offs were established statistically using a two-component Gaussian finite Mixture Model (GMM) using maximum likelihood estimation on a per-study basis as previously described [48]. The likelihood ratio (LR) tests were used to assess the prognostic information of one variable provided beyond that of another variable, assuming a χ2 distribution. Changes in the LR values (ΔLR-χ2) measured quantitatively the relative amount of information of one model compared with another. All statistical tests were two-sided at the 5% level of significance. Statistical analysis was conducted using the survival package (version 3.20, 24 August 2021) in the R software (version 3.5.2; 20 December 2018; http://www.cran.r-project.org/, accessed on 9 February 2022).

## 5. Conclusions

Our results showed the heterogeneity of *aCTCs* that can be observed in cancer patients and the interest in studying it. Using a *real-life* cohort of advanced mBC patients (Marseille, France), we highlighted the *g-aCTCs* subset as an independent prognostic factor in mBC patients, both regarding PFS and OS. This prognostic value might be further improved using the EMT status of this subset. The correlation of *aCTC* subsets, and notably of *g-aCTC*, with clinical and molecular data is new, but deserves further validation in larger cohorts. Our study suggests at least two clinical added values for enumerating *g-aCTCs*. It can help oncologists to identify which patients, after several lines of systemic therapy, might benefit from best supportive care alone. It can also serve as a tool for better prognostic stratification in early clinical trials testing novel therapeutics, which frequently enroll late-stage, already heavily treated and poorly responding patients.

## Figures and Tables

**Figure 1 cancers-14-00932-f001:**
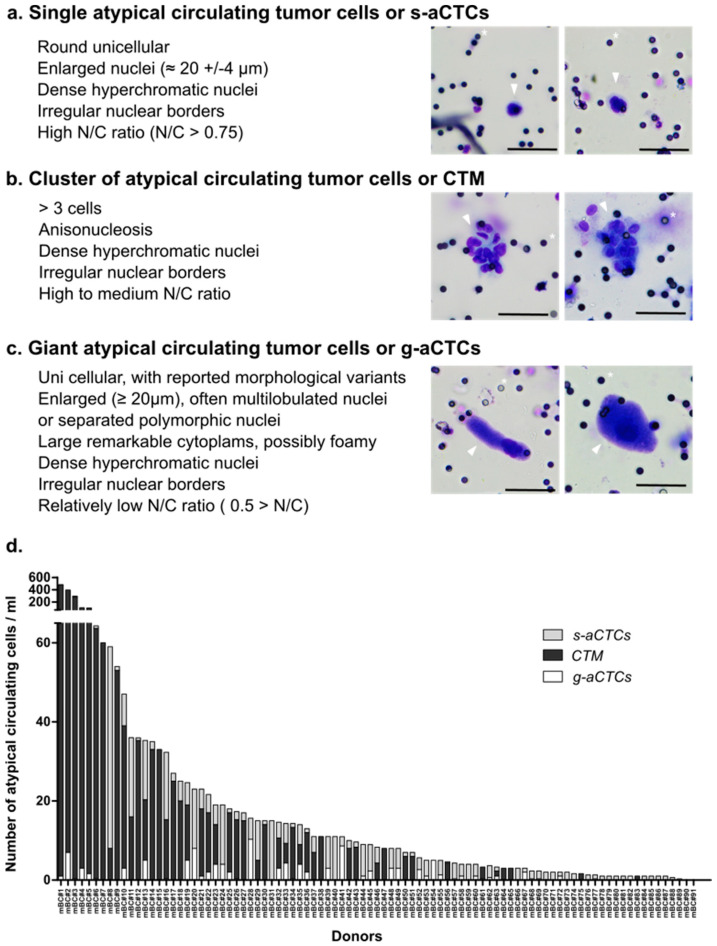
Images of atypical circulating cells and respective subsets’ distribution in mBC patients. (**a**–**c**) May–Grünwald staining of *aCTCs* isolated from the blood of mBC patients using the Screencell^®^Cyto device: (**a**) single cell or *s-aCTCs*; (**b**) cluster of cells or *CTM*; (**c**) giant cells or *g-aCTCs*. Small black dots are filter’s pores, marked with a white asterisk. Cells of interest are marked with an arrow. Scale represents 50 µm; (**d**) distribution of aCTC subsets in mBC patients (*n* = 91). Number of *s-aCTC*, *CTM* and *g-aCTC* are indicated per mL of blood.

**Figure 2 cancers-14-00932-f002:**
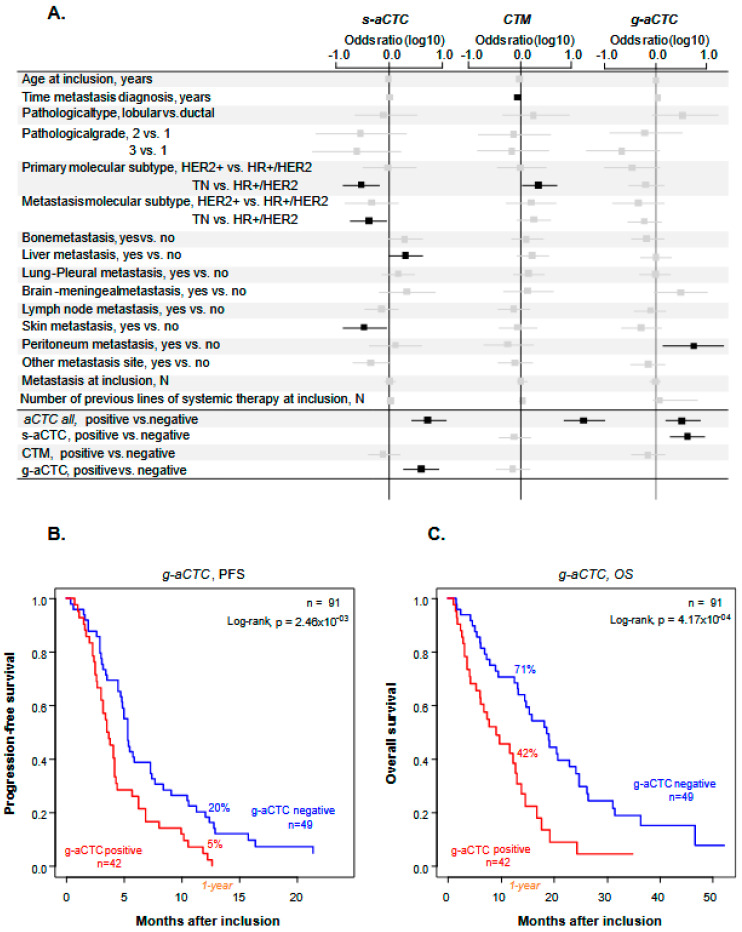
Correlations between *aCTC* subsets, clinicopathological variables and survival data. (**A**) Forest plot representation of the correlation between the total number of atypical cells detected or individual atypical cell subsets and clinical data (*n* = 91 patients). Odds ratios are indicated with confidence intervals (horizontal lines). Statistically significant data are in black, statistically insignificant data (crossing 0, vertical line) are in grey; (**B**,**C**) Kaplan–Meier PFS (**B**) and OS (**C**) curves of mBC patients according to the presence of *g-aCTCs.* Data are represented as mean ± SEM (*n* = 3). Differences were considered significant at *p* < 0.05. TN: triple-negative, HR+: Hormone Receptive-positive, HER2−/+: HER2 negative/positive.

**Figure 3 cancers-14-00932-f003:**
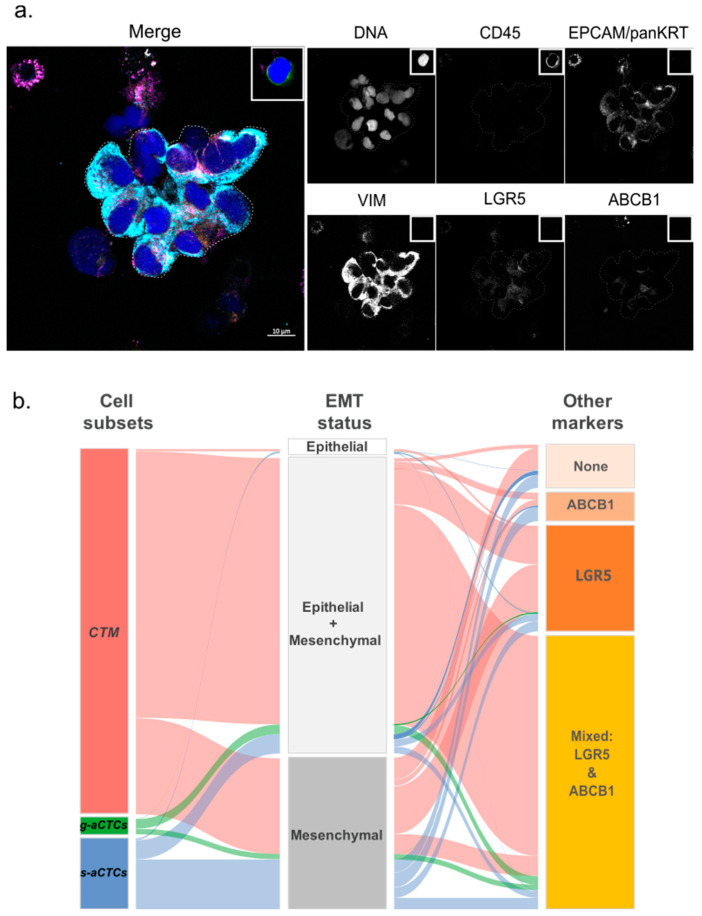
Example of the immunofluorescence staining of *CTM* isolated on ScreenCell^®^ filters. (**a**) Immunostaining of a *CTM* immobilized on the filter. This *CTM* shows a hybrid EMT status (EPCAM+, pan-KRT+, and VIM+) and weak expression of LGR5 and ABCB1. The staining obtained for a leukocyte (CD45+) is represented in the insert (top right) of each channel. Scale bar represents 10µm; (**b**) alluvial plot representation of the correlations between aCTC subsets and molecular markers. The graph shows the combined expression of EMT, stem-like and efflux pump markers for the three subsets of atypical cells (*CTM, g-aCTCs*, and *s-aCTCs*). The height of the blocks represents the size of the population: The thickness of a stream represents the number of cells contained in blocks interconnected by the stream.

**Table 1 cancers-14-00932-t001:** Clinical data, *aCTCs* and *aCTCs* subsets status (negative vs. positive cut-off), and survival data (PFS and OS).

Patients’ Characteristics		N (%)
Age at inclusion, years		55 (27–79)
Metastasis to diagnosis interval, years		3 (0.59–23)
Pathological type of primary tumor	ductal	75 (93%)
	lobular	6 (7%)
	missing	10
Pathological grade of primary	1	5 (7%)
	2	32 (43%)
	3	37 (50%)
	missing	17
Molecular subtype of primary tumor	HR+/HER2−	50 (56%)
	HER2+	11 (12%)
	TN	29 (32%)
	missing	1
Molecular subtype of metastasis	HR+/HER2−	36 (41%)
	HER2+	11 (13%)
	TN	40 (46%)
	missing	4
Bone metastasis	no	32 (35%)
	yes	59 (65%)
Liver metastasis	no	39 (43%)
	yes	52 (57%)
Lung-pleural metastasis	no	45 (49%)
	yes	46 (51%)
Brain-meningeal metastasis	no	81 (89%)
	yes	10 (11%)
Lymph node metastasis	no	34 (37%)
	yes	57 (63%)
Skin metastasis	no	73 (80%)
	yes	18 (20%)
Peritoneum metastasis	no	81 (89%)
	yes	10 (11%)
Other metastatic site	no	66 (73%)
	yes	25 (27%)
Number of metastasic sites at inclusion, N		3 (1–8)
Number of previous lines of systemic therapy at inclusion, N		4 (1–10)
Atypical circulating cells (all subsets)	Negative	35 (38%)
	Positive	56 (62%)
*s-aCTC*	Negative	42 (46%)
	Positive	49 (54%)
*CTM*	Negative	44 (48%)
	Positive	47 (52%)
*g-aCTC*	Negative	49 (54%)
	Positive	42 (46%)
Follow-up median, months (range)		12 (1–52)
PFS events, N (%)		90 (99%)
Median PFS, months (min–max)		5 (1–47)
1-year PFS, % [95% CI]		13% (8–22)
OS events, N (%)		68 (75%)
Median OS, months (min-max)		14 (1–52)
1-year OS, % [95% CI]		58% (49–70)

**Table 2 cancers-14-00932-t002:** Univariate and multivariate analyses of PFS and OS.

		PFS						OS					
		Univariate			Multivariate				Univariate		Multivariate		
		N	HR [95% CI]	*p*-Value	N	HR [95% CI]	*p*-Value	N	HR [95% CI]	*p*-Value	N	HR [95% CI]	*p*-Value
Age at inclusion ^#^, years		91	1.02 (1.002–1.04)	**0.0334**	91	1.02 (1.00–1.04)	**0.0378**	91	1.01 (0.99–1.04)	0.286			
Metastasis to diagnosis interval ^#^, years		91	1.02 (0.98–1.06)	0.34				91	1.03 (0.99–1.08)	0.173			
Pathological type of primary tumor	lobular vs. ductal	81	1.24 (0.53–2.86)	0.621				81	1.60 (0.57–4.49)	0.371			
Pathological grade of primary	2 vs. 1	74	0.74 (0.28–1.97)	0.453				74	0.64 (0.19–2.18)	0.533			
	3 vs. 1		0.59 (0.23–1.55)						0.53 (0.16–1.80)				
Molecular subtype of primary	HER2pos	90	1.47 (0.74–2.91)	0.151				90	1.50 (0.67–3.39)	0.262			
	TN		1.99 (0.96–4.12)						1.98 (0.85–4.64)				
Molecular subtype of metastasis	HER2pos	87	0.87 (0.43–1.76)	0.866				87	1.26 (0.58–2.73)	0.694			
	TN		1.05 (0.67–1.67)						1.25 (0.73–2.14)				
Bone metastasis	yes vs. no	91	0.89 (0.57–1.38)	0.591				91	1.04 (0.62–1.74)	0.892			
Liver metastasis	yes vs. no	91	0.81 (0.52–1.25)	0.335				91	1.01 (0.62–1.66)	0.958			
Lung-pleural metastasis	yes vs. no	91	1.09 (0.72–1.66)	0.679				91	1.06 (0.65–1.71)	0.817			
Brain-meningeal metastasis	yes vs. no	91	1.42 (0.73–2.75)	0.304				91	3.15 (1.58–6.28)	**0.00115**	91	2.74 (1.37–5.48)	**0.00454**
Lymph node metastasis	yes vs. no	91	1.06 (0.68–1.65)	0.784				91	1.14 (0.68–1.91)	0.607			
Skin metastasis	yes vs. no	91	1.02 (0.60–1.74)	0.942				91	1.43 (0.81–2.51)	0.219			
Peritoneum metastasis	yes vs. no	91	1.20 (0.62–2.32)	0.597				91	1.50 (0.71–3.16)	0.287			
Other metastatic site	yes vs. no	91	0.95 (0.59–1.53)	0.845				91	1.34 (0.80–2.23)	0.269			
Number of metastasic sites at inclusion ^#^, N		91	1.00 (0.86–1.15)	0.948				91	1.12 (0.96–1.30)	0.158			
Number of previous lines of systemic therapy at inclusion ^#^, N		91	1.10 (1.01–1.21)	**0.0348**	91	1.06 (0.96–1.16)	0.23193	91	1.11 (1.01–1.23)	**0.0396**	91	1.09 (0.98–1.20)	0.10355
all atypical circulating cells	positive vs. negative	91	1.37 (0.89–2.10)	0.158				91	1.50 (0.91–2.49)	0.115			
*s-aCTC*	positive vs. negative	91	1.51 (0.99–2.29)	0.056				91	1.51 (0.93–2.45)	0.093			
*CTM*	positive vs. negative	91	1.16 (0.77–1.76)	0.478				91	1.35 (0.83–2.20)	0.221			
*g-aCTC*	positive vs. negative	91	1.94 (1.25–3.01)	**0.00298**	91	1.87 (1.19–2.95)	**0.00661**	91	2.46 (1.47–4.12)	**0.000584**	91	2.23 (1.31–3.78)	**0.00298**

^#^: variables tested as continuous values. Statistically significant *p*-value are in bold.

## Data Availability

The datasets generated during and/or analyzed during the current study are available from the corresponding author on reasonable request. All patients’ data were anonymized and will be kept as such. Correspondence and requests for materials should be addressed to E.M.

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
