# Peer review of "Identification of Atypical Circulating Tumor Cells with Prognostic Value in Metastatic Breast Cancer Patients"

_cancers, 2022, doi:10.3390/cancers14040932_

Round 1
Reviewer 1 Report
In their manuscript “Identification of Atypical Circulating Tumor Cells with Prognostic Value in Metastatic Breast Cancer Patients”, Lopresti et al. analyzed the heterogeneity of CTCs and their potential use as biomarkers in metastatic breast cancer patients. It seems a very well-written manuscript. The highlight of the study is the use of a second-generation “filtration-based” technology, the Screencell, to isolate and analyze the atypical CTCs from the blood of metastatic breast cancer at the cytomorphological and molecular levels and their potential use as prognostic factor. The clinical significance of this is high because, in the perspective, it can help in clinical decisions for any systemic treatment or not.
Author Response
We thank the reviewer for her/his appreciation and comments.

Reviewer 2 Report
The current study has isolated and analyzed atypical cells found in the blood of metastatic breast cancer patients using a micro-filtration technic. They have identified three subsets of atypical circulating tumor cells (aCTCs) that can be used as a prognostic marker for progression-free and overall survivals. The article reveals crucial information for oncologists and cancer researchers which can be useful in determining advanced stages of breast cancer. There are a few major points that need to be addressed in the manuscript.
- The sample size (91) used in the study is relatively small to draw such a strong conclusion. It is recommended to include more patient samples.
- Why authors have chosen only metastatic breast cancer patients to conduct the study.
- What are the levels of aCTCs in primary tumor subtypes? If any, mention any published studies about circulating tumor cells in primary breast cancer.
- It is advisable to further analyze the isolated CTCs with additional markers to determine their phenotype and importance.
- Was the treatment regimen of the patients taken into consideration while evaluating the aCTCs? Authors should take into consideration the type of chemotherapeutics given to these patients to treat their primary tumors.
- Have authors considered any other factors which might contribute to the type and distribution of aCTCs in these patients?
Author Response
Reviewer#2
The current study has isolated and analyzed atypical cells found in the blood of metastatic breast cancer patients using a micro-filtration technic. They have identified three subsets of atypical circulating tumor cells (aCTCs) that can be used as a prognostic marker for progression-free and overall survivals. The article reveals crucial information for oncologists and cancer researchers, which can be useful in determining advanced stages of breast cancer.
We thank the reviewer for her/his appreciation and comments.
There are a few major points that need to be addressed in the manuscript.
- The sample size (n = 91) used in the study is relatively small to draw such a strong conclusion. It is recommended to include more patient samples.
We agree with Reviewer#2’s statement, which is why we were already very cautious in drawing our initial conclusion, as mentioned several times through the text and in the discussion section, more specifically in the 3.3 Limitation section.
Line 159: “Even if these results must be interpreted with caution given the high number of statistical tests performed for a medium-size cohort, interesting correlations were nonetheless observed”.
Line 380: “However, the molecular part of our study in 23 patients is still very preliminary and warrants future validation in larger series. For now, it should be regarded only as hypothesis-generating.”
Line 395: “Ideally, similar analyses should be reapplied to larger and more homogeneous series of patients.”
We have now taken additional precautions, and added the following sentence in the conclusion section as well.
line 422 “The correlation of aCTCs subsets, and notably of g-aCTC, with clinical and molecular data is new, but deserves further validation in larger cohorts.”
We agree that more samples would have been better. Unfortunately, the study is closed since September 2019 and we do not have access to new samples with the same characteristics.
https://clinicaltrials.gov/ct2/show/NCT02342158
That said, instead of increasing the size of the cohort to compensate for the large range of prior systemic therapies and the heterogeneity in term of molecular subtypes of the cohort, it may be more productive, in the future, we will try to analyze more homogeneous cohorts to address specific questions.
Why authors have chosen only metastatic breast cancer patients to conduct the study.
We chose to work on metastatic breast cancer patients based on data from the literature showing that CTCs are more abundant in mBC. This seemed an important point to study heterogeneity among CTCs. A new cohort on neo-adjuvant non-metastatic patients will start in 2022 in our Institute. We will do a similar analysis.
- What are the levels of aCTCs in primary tumor subtypes? If any, mention any published studies about circulating tumor cells in primary breast cancer.
In our hand, we do not have any experience of aCTCs in primary tumor samples with the Screencell™ assay. The laboratory of Dr N Aceto has recently published an article showing that the mean number of CTCs in primary tumors is of: 3 CTCs, but the total volume of blood analyzed is not specify (ranging from 5 to 10 mL of blood) (DOI: 10.1038/s41416-021-01327-8). Most studies mention the percentage of positivity rather than the mean number of CTCs par mL. In a previous multicentric assay in neoadjuvant non-metastatic inflammatory breast cancer patients, the median CTC count with CellSearch System for positive cases was 3.5 (range [1–559]), with 55 patients (39%) having detectable CTCs (DOI: 10.1093/annonc/mdw535). Of note that inflammatory BC represent a population with aggressive disease, and high metastatic risk. Altogether, data about the number of aCTCs in primary tumor subtypes are scarce.
We have now mentioned this line 308
- It is advisable to further analyze the isolated CTCs with additional markers to determine their phenotype and importance.
We approve this advice and, in the future, we will design studies so that we can provide a more exhaustive analysis of aCTCs inference, using more markers or including in-depth molecular analyses. Such an approach might also provide additional and more refined prognostic value than the single count of all pooled CTCs types.
This is now mentioned in the discussion section line 363
“Both hypotheses … require further investigations to clarify the origin of these giant cells. This would help refine their role as biomarkers, including in early disease »
- Was the treatment regimen of the patients taken into consideration while evaluating the aCTCs?
Information about treatment regimen has been considered, and 91% of patients had previously received taxane and/or anthracycline treatments since the diagnosis of metastasis. The systemic treatments received before and after inclusion are summarized in Table S1; as shown, the number of lines of treatment received by each patient was too heterogeneous to be analyzed deeper from our cohort of 91 cases.
We have mentioned this line 330: “Our cohort having received multiple prior systemic therapies, one can hypothesize that the predictive value is less powerful in terminal disease »
Authors should take into consideration the type of chemotherapeutics given to these patients to treat their primary tumors.
All our patients had a metastatic disease and the three different molecular subtypes of breast cancer were represented (HR+/HER2-; HER2+, and TN). Thus, the adjuvant systemic treatments that our patients had received are likely heterogeneous, both because of different molecular subtypes, but also because of possible comorbidities and patients’ refusal. Furthermore, this information about adjuvant treatment was not always available since some patients had been previously treated outside our institution some years ago. However, we can indeed consider that either the adjuvant systemic treatment or the tumor subtypes, as well as many other variables, influenced the number of aCTCs and aCTCs subsets (Figure 3a). For example, a number of CTM above the cut-off was more frequent in TN patients, whereas aCTCs positivity was higher in HER2+ tumors. Whether this is due to the subtype or the previously received regimen is impossible to say with this cohort, and remains a challenging issue in term of study design.
- Have authors considered any other factors which might contribute to the type and distribution of aCTCs in these patients?
Indeed, many other factors might contribute to the type and distribution of aCTCs in the patients. For example, one factor that we can think of is the recognition of specific cell subset by immune cells. Indeed, it has been suggested that clusters of cells were less susceptible to anti-tumor immunity. Cell fusion might also be a cancer cell strategy used to avoid anti-tumor immunity. As mentioned previously, more homogeneous cohorts should help answering such question.
We have now suggested this line 364: “Both hypotheses, fusion or myeloid origin, require further investigations to clarify the origin and the function of these giant cells”
Overall, we thank the reviewer for her/his comments and are sorry not to be able to analyze additional samples.

Reviewer 3 Report
This study is a needed, and interesting investigation of the heterogeneity of Circulating breast cancer cells.
My main concerns about the manuscript are related to the presentation and language:
The readability of the text is generally low, I would suggest the authors to consult a professional scientific english language editor. For example, many text sections are full of brackets, which reduces the quality of the language. Some text sentences and sections contain more brackets than normal text. Also, the text is not written in an accessible manner, some sentences span over 7 or more rows. For example, two sentences in the conclusion section contain " i) ", and ii ), which not only are repetitions but significantly reduced readability. The Fig 1d should be wider. None of the figure legends have the correct format, they are much too long. In particular, Fig 3 figure legend, which is almost half a page long, contains information on methods and procedures that should not be part of a figure legend.
The quality of the scientific English language is generally low, the readability is low, and it would be better to not mention details on methods anywhere else than in the M&M section.
Author Response
"Please see the attachment"

Reviewer 4 Report
The manuscript details a well-designed important clinical study about the characterization of several subsets of CTCs and their role in disease progression. The paper is well-written and easy to read.
The 6-colour immunostaining is very useful for the identification of different subgroups.
The main finding of the study that g-aCTCs subset is an independent prognostic factor, is novel and important for future diagnostics and therapeutics.
Author Response
"Please see the attachment."

Round 2
Reviewer 2 Report
The manuscript can be accepted in the present form.